# Platelet-derived growth factor (PDGF) signaling directs cardiomyocyte movement toward the midline during heart tube assembly

Joshua Bloomekatz[1], Reena Singh[2,3], Owen WJ Prall[2,4], Ariel C Dunn[1], Megan Vaughan[1], Chin-San Loo[1], Richard P Harvey[2,3,5]*, Deborah Yelon[1]*

[1]Division of Biological Sciences, University of California, San Diego, La Jolla, United States; [2]Victor Chang Cardiac Research Institute, Darlinghurst, Australia; [3]St. Vincent's Clinical School, University of New South Wales, Kensington, Australia; [4]Peter MacCallum Cancer Centre, Melbourne, Australia; [5]School of Biotechnology and Biomolecular Science, University of New South Wales, Kensington, Australia

**Abstract** Communication between neighboring tissues plays a central role in guiding organ morphogenesis. During heart tube assembly, interactions with the adjacent endoderm control the medial movement of cardiomyocytes, a process referred to as cardiac fusion. However, the molecular underpinnings of this endodermal-myocardial relationship remain unclear. Here, we show an essential role for platelet-derived growth factor receptor alpha (Pdgfra) in directing cardiac fusion. Mutation of *pdgfra* disrupts heart tube assembly in both zebrafish and mouse. Timelapse analysis of individual cardiomyocyte trajectories reveals misdirected cells in zebrafish *pdgfra* mutants, suggesting that PDGF signaling steers cardiomyocytes toward the midline during cardiac fusion. Intriguingly, the ligand *pdgfaa* is expressed in the endoderm medial to the *pdgfra*-expressing myocardial precursors. Ectopic expression of *pdgfaa* interferes with cardiac fusion, consistent with an instructive role for PDGF signaling. Together, these data uncover a novel mechanism through which endodermal-myocardial communication can guide the cell movements that initiate cardiac morphogenesis.

*For correspondence: r.harvey@ victorchang.edu.au (RPH); dyelon@ucsd.edu (DY)

## Introduction

Organogenesis relies upon the coordinated regulation of precisely defined patterns of cell movement. Multiple precursor cell populations must convene at the appropriate location and organize into the correct configuration in order to insure proper organ function. Differential adhesion and paracrine signaling between neighboring tissues often influence the specific routes traveled by precursor cells during morphogenesis (*Scarpa and Mayor, 2016*). However, the molecular mechanisms through which tissue interactions guide organ assembly remain poorly understood.

Heart formation requires the coordinated movement of myocardial precursor cells from their bilateral origins toward the embryonic midline, where they meet and merge through a process called cardiac fusion (*Evans et al., 2010*). Cardiac fusion is essential for the construction of the heart tube, which provides a fundamental foundation for subsequent steps in cardiac morphogenesis. During cardiac fusion, the medial movement of the myocardium is considered to be a collective cell behavior: the cardiomyocytes travel along relatively parallel paths with very little neighbor exchange (*Holtzman et al., 2007*) and simultaneously form intercellular junctions and create a primitive epithelial sheet (*Linask, 1992*; *Manasek, 1968*; *Stainier et al., 1993*; *Trinh and Stainier, 2004*; *Ye et al.,*

**eLife digest** In the growing embryo, the heart initially develops in the form of a simple tube. Its outer layer is made up of muscular cells, called myocardial cells, that pump blood through the tube. Before the heart tube develops, two groups of myocardial cells exist – one on each side of the embryo. To assemble the heart, these two populations of cells must move as a group to the middle of the embryo, where they meet and merge through a process called cardiac fusion. This movement of myocardial cells toward the middle of the embryo depends upon interactions with a neighboring tissue called the endoderm. How the endoderm directs the movement of the myocardial cells was not well understood.

The PDGF signaling pathway guides the movement of several different types of cells in the body, but it had not been previously linked to the early stages of heart tube assembly. In this pathway, a molecule called platelet-derived growth factor (PDGF) binds to PDGF receptors that sit on the surface of cells. Using microscopy and genetic analysis to study zebrafish and mouse embryos, Bloomekatz et al. now show that embryos that carry mutations in a gene that encodes a PDGF receptor suffer from defects in heart tube assembly. Further examination of the mutant zebrafish embryos revealed that the myocardial cells were not properly directed toward the middle of the embryo. In fact, many of these cells appeared to move away from the midline.

Bloomekatz et al. also observed that, in normal embryos, the endoderm cells that lie adjacent to the myocardial cells produce PDGF. Therefore, it appears that PDGF produced by the endoderm could interact with PDGF receptors on the myocardial cells to direct these cells toward the middle of the embryo. The next step will be to figure out how this signaling influences the machinery inside the myocardial cells that controls their movement. Ultimately, this knowledge could lead to new ways to identify and treat congenital heart diseases.

*2015*). Whether these coherent patterns of myocardial movement reflect active migration or passive morphogenesis is not yet resolved (*Aleksandrova et al., 2015*; *Dehaan, 1963*; *Varner and Taber, 2012*; *Xie et al., 2016*; *Ye et al., 2015*). In either case, it is important to elucidate the specific signals that dictate the medial direction of myocardial trajectories during cardiac fusion.

Several lines of evidence indicate that cardiac fusion is mediated by interactions between the myocardium and the adjacent anterior endoderm. In both mouse and zebrafish, mutations that block endoderm formation or disrupt endoderm integrity also inhibit cardiac fusion (*Alexander et al., 1999*; *Holtzman et al., 2007*; *Kawahara et al., 2009*; *Kikuchi et al., 2001*; *Kupperman et al., 2000*; *Li et al., 2004*; *Mendelson et al., 2015*; *Molkentin et al., 1997*; *Osborne et al., 2008*; *Ragkousi et al., 2011*; *Roebroek et al., 1998*; *Ye and Lin, 2013*; *Yelon et al., 1999*). Studies tracking both endodermal and myocardial movement in chick have suggested that endodermal contraction provides a physical force that pulls the myocardium toward the midline (*Aleksandrova et al., 2015*; *Cui et al., 2009*; *Varner and Taber, 2012*). However, while endodermal forces may influence initial phases of cardiac fusion, the observed patterns of endoderm behavior seem insufficient to account for the entire path traversed by the moving cardiomyocytes (*Aleksandrova et al., 2015*; *Cui et al., 2009*; *Varner and Taber, 2012*; *Xie et al., 2016*; *Ye et al., 2015*). Moreover, observations of myocardial cell protrusions have suggested that these cells may actively migrate in response to endodermal cues (*Dehaan, 1963*; *Haack et al., 2014*; *Ye et al., 2015*). While it is clear that the endoderm plays an important role in facilitating cardiac fusion, the molecular underpinnings of the endodermal-myocardial relationship are still unknown.

Here, we reveal a novel connection between the endoderm and myocardium by discovering a new role for platelet-derived growth factor (PDGF) signaling. PDGFs signal through receptor tyrosine kinases and are well known for their mitogenic activity (*Andrae et al., 2008*), as well as for their role in guiding the migration of mesenchymal cells (*Ataliotis et al., 1995*; *Yang et al., 2008*). However, PDGF signaling has not been previously implicated in heart tube assembly, even though it is known to be important for later aspects of heart development, such as the contribution of cardiac neural crest cells to the outflow tract (*Morrison-Graham et al., 1992*; *Schatteman et al., 1995*;

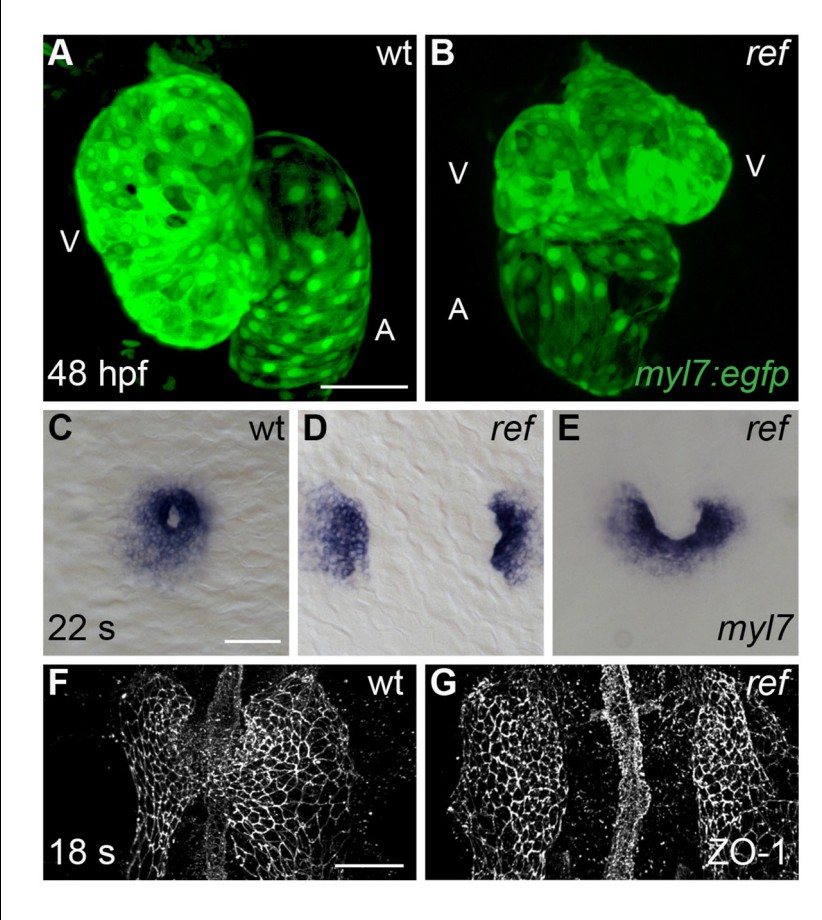

**Figure 1.** Cardiac fusion defects in *refuse-to-fuse* (*ref*) mutants. (**A,B**) Three-dimensional reconstructions depict wild-type (wt) and *ref* mutant hearts expressing the myocardial reporter transgene *Tg(myl7:egfp)* at 48 hpf. In contrast to the normal contours of the wt heart (**A**), the *ref* mutant heart (**B**) often displays a bifurcated, two-lobed ventricle and a misshapen atrium. See *Figure 1—figure supplement 1* and *Table 1* for more information on the range of cardiac phenotypes observed in *ref* mutants at 48 hpf. A: atrium; V: ventricle. (**C–G**) Dorsal (**C–E**) and ventral (**F,G**) views, anterior to the top, of wt (**C,F**) and *ref* (**D,E,G**) mutant embryos displaying the expression of *myl7* at the 22 somite stage (22 s; **C–E**) and the localization of ZO-1 at the 18 somite stage (18 s; **F,G**). (**C–E**) In *ref* mutants at this stage, cardiomyocytes typically fail to fuse at the midline (**D**) or only fuse posteriorly (**E**). Note that the *ref* mutation is incompletely penetrant, although its penetrance is more evident at 20 s than at 48 hpf (*Table 2*), suggesting that some *ref* mutants recover as development proceeds. (**F,G**) ZO-1 localization highlights junctions forming within the maturing epithelium of the ALPM in both wt and *ref* mutant embryos. The ventral portion of the neural tube located at the midline is also visible. By 18 s, the wt ALPM (**F**) has initiated fusion at the midline, whereas the two sides of the *ref* mutant ALPM are still separate (**G**). Scale bars: 60 µm.

The following figure supplements are available for figure 1:

**Figure supplement 1.** *ref* mutant embryos display a range of cardiac defects.

**Figure supplement 2.** The anterior endoderm forms normally in *ref* mutant embryos.

**Figure supplement 3.** Endocardial cells move to the midline in *ref* mutant embryos.

*Tallquist and Soriano, 2003*), the formation of the inflow tract (*Bleyl et al., 2010*), and the formation of epicardial derivatives (*Smith et al., 2011*).

Our analysis of early morphogenetic defects caused by mutation of the gene encoding PDGF receptor alpha (Pdgfra) uncovers an essential function for Pdgfra during cardiac fusion in both

**Table 1.** Variable expressivity of cardiac phenotype in *ref* mutant embryos.

| Stage | Phenotype | Number of embryos |
|---|---|---|
| 22 s[*] | | |
| | No fusion | 8 |
| | Only posterior fusion | 7 |
| 48 hpf[†] | | |
| | Bifurcated ventricle | 229 |
| | Shrunken heart | 147 |
| | Abnormal looping | 74 |
| | Cardia bifida | 4 |

[*] Tabulated from 15 embryos with morphologically evident phenotypes from clutch depicted in **Figure 1C–E** and in **Table 2**. See **Figure 2—figure supplement 2** for additional data.

[†] Tabulated from 454 embryos with morphologically evident phenotypes collected from multiple clutches. See **Figure 1—figure supplement 1** for representative images.

zebrafish and mouse. Notably, through live imaging of individual cell movements in zebrafish mutants, we find that *pdgfra* is crucial for guiding cardiomyocyte movement toward the midline. Furthermore, our studies suggest that expression of PDGF ligands by the anterior endoderm could facilitate interaction of this tissue with the *pdgfra*-expressing myocardial precursors. Thus, our work supports a model in which PDGF signaling underlies communication between the endoderm and myocardium and thereby directs the cell movements that initiate heart tube assembly. These insights into the regulation of cardiomyocyte behavior provide new ideas regarding the etiology of diseases associated with aberrant cell movement (*Friedl and Gilmour, 2009*), including congenital heart diseases (CHDs) caused by defective myocardial morphogenesis (*Bleyl et al., 2010*; *Briggs et al., 2012*; *Neeb et al., 2013*; *Samsa et al., 2013*).

## Results

### *refuse-to-fuse (ref)* mutants display cardiac fusion defects

In a screen for ethylnitrosourea-induced mutations that disrupt cardiac morphogenesis in zebrafish (*Auman et al., 2007*), we identified a recessive lethal mutation, *refuse-to-fuse* (*ref*), that causes abnormal cardiac chamber morphology. Instead of the normal curvatures of the wild-type ventricle (*Figure 1A*), *ref* mutants often displayed a bifurcated ventricle at 48 hours post-fertilization (hpf) (*Figure 1B*). This phenotype was the most common among a range of cardiac defects in *ref* mutants (*Figure 1—figure supplement 1*; *Table 1*). On rare occasions, we found *ref* mutants with cardia bifida, a condition in which two separate hearts form in lateral positions (*Table 1*). The observed bifurcated ventricle and cardia bifida phenotypes led us to hypothesize that the *ref* mutation might interfere with cardiac fusion. In wild-type embryos, cardiac fusion results in the formation of a ring of cardiomyocytes at the midline by the 22 somite stage (*Figure 1C*). In contrast, *ref* mutant cardiomyocytes failed to fuse into a ring and instead remained in separate bilateral domains (*Figure 1D*; *Table 1*) or fused only in posterior positions, creating a horseshoe shape (*Figure 1E*; *Table 1*). Similar fusion defects were also observed when examining a broader portion of the anterior lateral plate mesoderm (ALPM) encompassing the heart fields (*Figure 1F,G* and Figure 5A–F).

Since prior studies in zebrafish have shown that defects in endoderm specification or morphogenesis can inhibit cardiac fusion (*Alexander et al., 1999*; *Holtzman et al., 2007*; *Kawahara et al., 2009*; *Kikuchi et al., 2001*; *Kupperman et al., 2000*; *Mendelson et al., 2015*; *Osborne et al., 2008*; *Ye and Lin, 2013*; *Yelon et al., 1999*), we examined the status of the endoderm in *ref* mutants. During gastrulation stages, the specification and movement of endodermal cells appeared normal in *ref* mutants (*Figure 1—figure supplement 2A–D*). In addition, the differentiation and morphology of the anterior endoderm in *ref* mutants appeared intact during the stages when cardiac fusion takes place (*Figure 1—figure supplement 2E–H*). The normal appearance of the *ref* mutant

**Table 2.** Penetrance of cardiac phenotype in *ref* mutant embryos.

| Stage | Total # embryos | # +/+ embryos | # +/− embryos | # −/− embryos | # with evident cardiac defects | Approximate penetrance |
|---|---|---|---|---|---|---|
| 22 s | 61 | 14 | 27 | 20 | 15 | 75% |
| 48 hpf | 522 | NG[*] | NG[*] | NG[*] | 58 | 44%[†] |

[*] NG=not genotyped.
[†] Calculated with the assumption that 25% of embryos are -/-.

endoderm was consistent with the unaltered progress of the endocardial precursor cells in *ref* mutants: endocardial cells require interactions with the anterior endoderm for their medial movement during cardiac fusion (*Holtzman et al., 2007*; *Wong et al., 2012*; *Xie et al., 2016*), and the *ref* mutant endocardium seemed to reach the midline normally (*Figure 1—figure supplement 3*). Taken together, our data suggest that defects in myocardial movement, as opposed to defects in the endoderm, cause the bifurcated cardiac morphology in *ref* mutants.

## *ref* is a loss-of-function mutation in *pdgfra*

In order to identify the genomic lesion responsible for the *ref* mutant phenotype, we mapped the *ref* locus to a <0.1 cM region on linkage group 20 containing six annotated genes (*Figure 2A*). Our examination of *ref* mutant cDNA revealed missplicing in one of these genes, *platelet-derived growth factor receptor alpha (pdgfra)* (*Figure 2B*). Specifically, we noted that exon 14 was omitted or truncated in the *pdgfra* messages detected in *ref* mutant cDNA. Furthermore, *ref* mutant genomic DNA contained a G-to-A mutation in the first nucleotide of intron 15 of *pdgfra* (*Figure 2C,D*). Since a G at the exon/intron boundary is an essential conserved feature of splice sites, we infer that this mutation would disrupt *pdgfra* splicing. The misspliced *pdgfra* messages found in *ref* mutants cause a frameshift in the coding sequence, resulting in a premature truncation prior to the transmembrane domain of Pdgfra (*Figure 2D*). In concordance with the concept that premature stop codons often lead to nonsense-mediated decay, we detected a global reduction of *pdgfra* mRNA in *ref* mutants (*Figure 2E,F*).

We next compared the *ref* mutant phenotype to the effects of another mutation in *pdgfra*, *b1059*. The *b1059* allele is a missense mutation that disrupts a conserved residue within the tyrosine kinase domain of Pdgfra (*Eberhart et al., 2008*) (*Figure 2D*). Previous studies of *b1059* mutant embryos focused on their dorsal jaw defects (*Eberhart et al., 2008*); our analysis also uncovered dorsal jaw defects in *ref* mutants (*Figure 2—figure supplement 1B,E*), as well as cardiac fusion defects in *b1059* mutants (*Figure 2G,I*; *Figure 2—figure supplement 2*). Through complementation testing, we found that *ref* and *b1059* fail to complement each other; transheterozygotes displayed defects in both cardiac fusion (*Figure 2J*; *Figure 2—figure supplement 2*) and dorsal jaw formation (*Figure 2—figure supplement 1C,F*). Finally, we found that injection of a morpholino targeting *pdgfra* also interfered with cardiac fusion (*Figure 2H*; *Figure 2—figure supplement 2*). Together, our mapping, sequencing, complementation testing, and morpholino data support the conclusion that the *ref* mutation causes inappropriate splicing of *pdgfra*, resulting in diminished *pdgfra* function and cardiac fusion defects.

## Mutation of *Pdgfra* disrupts heart tube assembly in mice

Although prior work in mouse has revealed functions for PDGF signaling during later stages of heart development (*Grüneberg and Truslove, 1960*; *Richarte et al., 2007*; *Schatteman et al., 1995*), these studies did not report an earlier role for PDGFRα during cardiac fusion or heart tube assembly. In contrast, analysis of the *Patch* (*Ph*) mutant, carrying a chromosome deletion including *Pdgfra*, did reveal early cardiac phenotypes, as well as yolk sac defects (*Orr-Urtreger et al., 1992*). Exploration of the early functions of PDGFRα has been complicated by the variability of *Pdgfra* mutant phenotypes, due in part to genetic background (*Grüneberg and Truslove, 1960*; *Orr-Urtreger and Lonai, 1992*; *Schatteman et al., 1995*; *Soriano, 1997*; *Tallquist and Soriano, 2003*). Since the C57BL/6 background was reported to generate more severe phenotypes in the *Ph* mutant (*Orr-Urtreger et al., 1992*), we chose to analyze mouse embryos carrying *Pdgfra* null alleles on a C57BL/

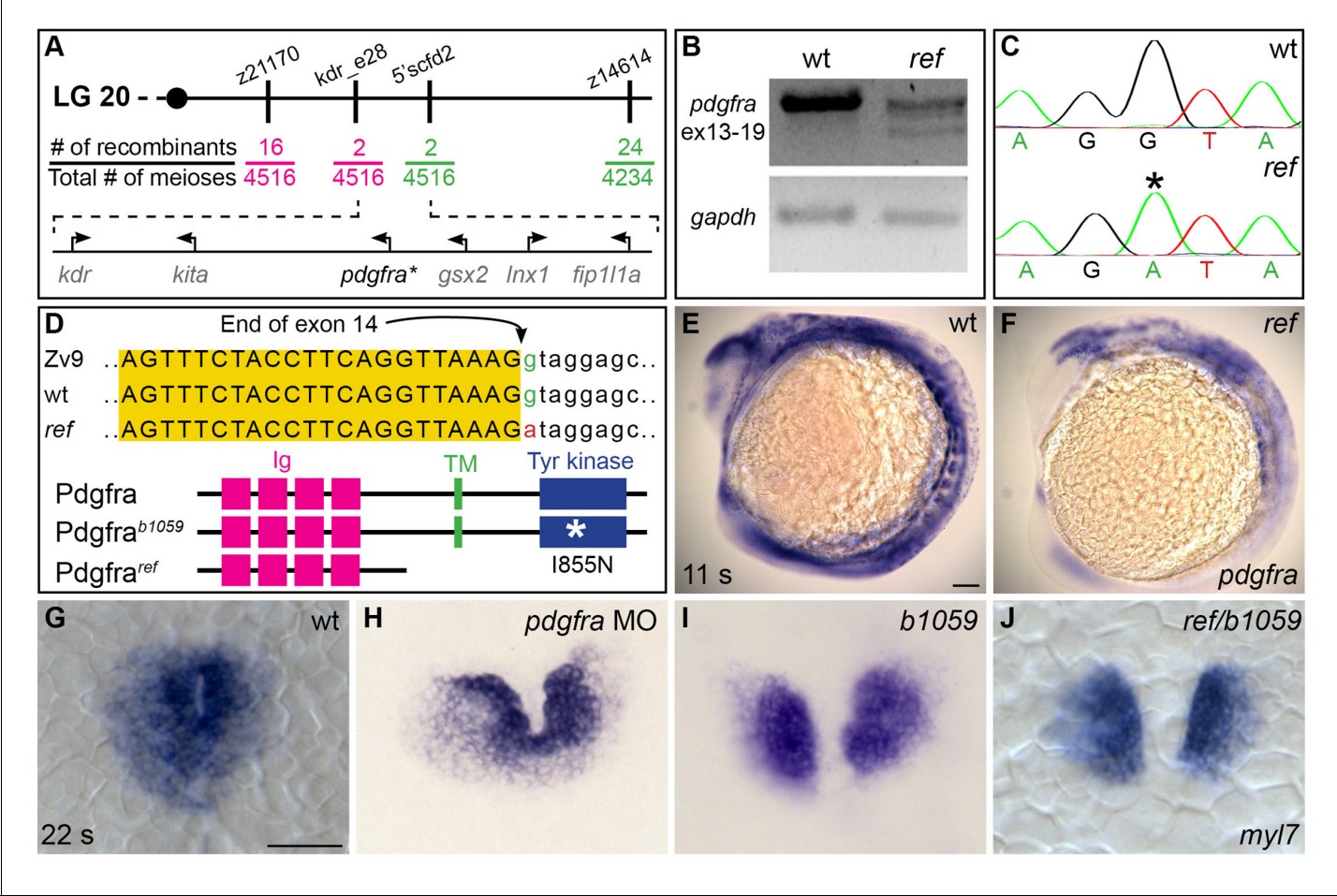

Figure 2. *ref* is a loss-of-function mutation in *pdgfra*. (A) Polymorphic markers (z21170, kdr_e28, 5'scfd2, z14614) were used to map meiotic recombination events, narrowing the region containing the *ref* mutation to <0.1 cM on linkage group (LG) 20. (See also *Table 3*). Fractions indicate frequencies of proximal (magenta) and distal (green) recombination between markers and *ref*. Six annotated genes are present in this region (GRCv10); sequence analysis of *kdr*, *kita*, *gsx2*, *lnx1*, and *fip1l1a* in *ref* mutants revealed only missense mutations that led to conserved amino acid changes. (B) RT-PCR spanning exons 13–19 of *pdgfra* generates a single, properly spliced product from homozygous wt embryos and multiple, smaller products from *ref* mutant embryos. Sequencing revealed that exon 14 was either omitted or truncated in these smaller products; in all cases, the observed missplicing would result in a frameshift followed by a premature stop codon. Although we did not detect any normally spliced *pdgfra* products in *ref* mutants, we cannot rule out the presence of low levels of wild-type mRNA. RT-PCR of *gapdh* demonstrates use of comparable amounts of template. (C,D) Sequencing the e14i15 exon-intron boundary of *pdgfra* revealed that *ref* mutant genomic DNA contains a G-to-A mutation in a conserved intronic nucleotide required for proper splicing. Chromatograms (C) and sequence alignment (D) show position of the mutation relative to reference sequences. Schematics (D) depict the proteins predicted to result from the wt, *ref,* and *b1059* alleles of *pdgfra*; immunoglobulin (magenta), transmembrane (green), and tyrosine kinase (blue) domains are shown. (E,F) Lateral views depict expression of *pdgfra* at 11 s. Expression levels are lower in *ref* mutants (F; n = 5/5) than in wt (E). (G–J) Dorsal views, anterior to the top, of *myl7* expression at 22 s. In contrast to wt (G), cardiac fusion defects are evident in embryos injected with a *pdgfra* morpholino (MO) (H), *b1059* homozygous mutant embryos (I), and *ref/b1059* transheterozygous mutant embryos (J). See *Figure 2—figure supplement 2* for additional information on the prevalence of each of these phenotypes. Scale bars: 60 μm.

The following figure supplements are available for figure 2:

**Figure supplement 1.** *ref* mutants display craniofacial defects.

**Figure supplement 2.** Comparison of cardiac fusion phenotypes resulting from alteration of PDGF signaling.

6J background at E9.5, using expression of *Nkx2-5* to highlight heart morphology (*Figure 3A–E'*). We note that we encountered *Pdgfra* null mutants on this background at E9.5 less often than predicted (*Table 4*), potentially because they fail to survive until this stage. Although the cause of this

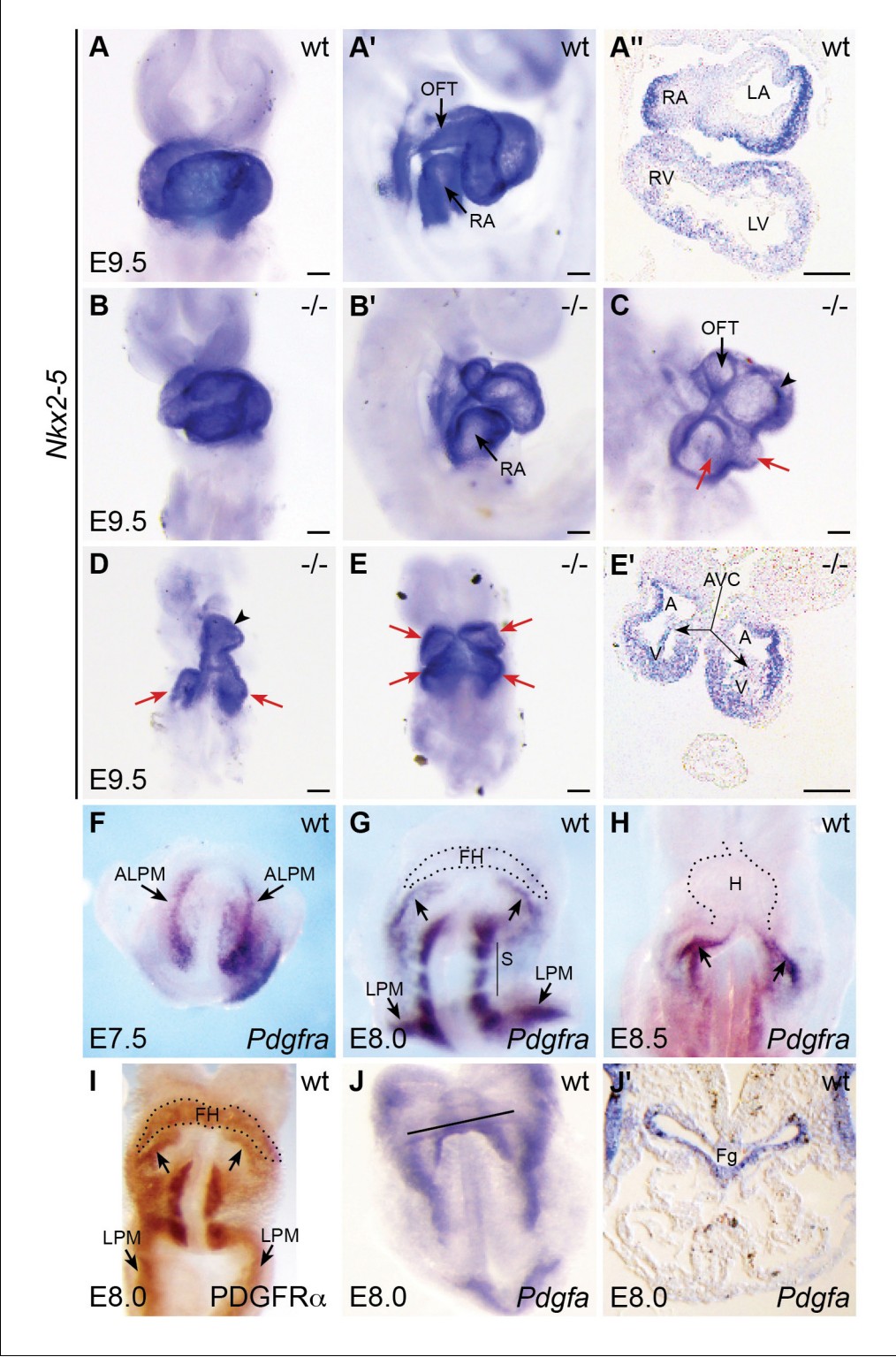

**Figure 3.** *Pdgfra* mouse mutants display defects in heart tube assembly. (**A–E'**) Ventral views (**A,B,D,E**), lateral views (**A',B',C**), and transverse sections (**A'',E'**) depict *Nkx2-5* expression in wt (**A–A''**) and *Pdgfra* homozygous mutant (**B–E'**) mice on a C57BL/6J background at E9.5. Some *Pdgfra* mutants display relatively mild cardiac defects (**B,B'**), including defects in heart tube rotation. Other *Pdgfra* mutants display severe cardiac defects that could result from hindered cardiac fusion (**C–E'**). These defects include incomplete heart tube assembly (**C,D**) with two inflow/common atrial regions (red arrows) and a single ventricle (black arrowhead) or with cardia bifida (**E,E'**;

*Figure 3 continued on next page*

*Figure 3 continued*

arrows indicate unfused ventricles and atria). See *Tables 4* and *5* for additional information on the prevalence of mutant phenotypes. A: atrium; AVC: atrioventricular canal; LA: left atrium; LV: left ventricle; OFT: outflow tract; RA: right atrium; RV: right ventricle; V: ventricle. Scale bars: 100 µm. (**F–I**) Ventral views depict localization of *Pdgfra* mRNA (**F–H**) and PDGFRα protein (**I**) in wt embryos from E7.5 to E8.5. *Pdgfra* expression is seen in anterior lateral plate mesoderm (ALPM) at E7.5 (**F**) and persists in the caudal region of the forming heart at E8.0 (**G**; arrows) and in the inflow tract at E8.5 (**H**; arrows). *Pdgfra* is also expressed in somites (S) and caudal lateral plate mesoderm (LPM) at E8.0-E8.5. Although mRNA levels are diminished, PDGFRα protein localization is maintained throughout the forming heart at E8.0 (**I**). FH: forming heart; H: heart. (**J,J'**) Ventral view (**J**) and transverse section (**J'**) depict the localization of *Pdgfa* mRNA at E8.0. Black line in **J** indicates plane of transverse section shown in **J'**. At this stage, *Pdgfa* expression is seen in the foregut endoderm (Fg) and lateral ectoderm.

---

loss remains to be identified, it is consistent with the expression of *Pdgfra* during early embryogenesis (e.g. *Artus et al., 2013*; *Palmieri et al., 1992*; *Schatteman et al., 1992*), and with studies in other vertebrates revealing a role for *Pdgfra* during gastrulation (*Nagel et al., 2004*; *Yang et al., 2008*).

Our analysis revealed a range of early defects in cardiac morphogenesis in homozygous *Pdgfra* mutants at E9.5 (*Figure 3A–E'*; *Table 5*). Wild-type hearts had undergone looping and exhibited distinct left and right atrial and ventricular chambers (*Figure 3A–A''*). Some *Pdgfra* homozygous mutant hearts displayed relatively mild defects in heart looping as well as in the size and shape of the cardiac chambers and their inflow and outflow tracts (*Figure 3B,B'*). Other *Pdgfra* mutant hearts displayed more severe disruptions that could be the consequence of abnormal cardiac fusion (*Figure 3C–E'*): the most prominent were embryos with a split inflow/common atrial region connected to a single ventricle (*Figure 3C,D*), and we also observed a single embryo with near total cardia bifida (*Figure 3E,E'*). *Pdgfra* mutants were often smaller than wild-type littermates, consistent with previous observations (*Orr-Urtreger et al., 1992*); however, the observed severe cardiac defects (C-E') are not likely a result of general developmental delay, as these mutant phenotypes do not resemble wild-type cardiac morphology at younger stages. Severely affected embryos had not turned, as previously observed (*Soriano, 1997*). We did not observe omphalocele as reported (*Soriano, 1997*), although these previous observations were made at later time points than examined here. The majority of *Pdgfra* mutants died by E10.5, slightly earlier than reported for the majority of *Ph* mutants (*Orr-Urtreger et al., 1992*). Altogether, our data uncover a previously unappreciated influence of *Pdgfra* on the early stages of cardiac morphogenesis in mice. In combination with the phenotype of *ref* mutants, these studies suggest that Pdgfra plays a conserved role in regulating heart tube assembly.

## *pdgfra* is expressed within the ALPM while cardiac fusion is underway

To further elucidate how Pdgfra influences heart tube assembly, we next examined the expression pattern of *pdgfra* during cardiac fusion in zebrafish. We found robust expression of *pdgfra* within the ALPM and in migrating neural crest cells (*Figure 4A–D*). The domains of *pdgfra* expression in the ALPM matched those of *hand2* (*Figure 4E–J*), which is expressed in the territories that contain myocardial precursor cells and is excluded from the territories containing endocardial precursors (*Schoenebeck et al., 2007*). As cardiac fusion proceeds, *hand2* continues to be expressed in the

---

**Table 3.** Primers used to map recombinants.

| Marker | Forward primer | Reverse primer |
| --- | --- | --- |
| 5'SCFD2 | CGCGTTACCAGAGAGACACA | TTCTCGGCAGGATAAATTGG |
| Z14614 | AAACACATGCACAATGGTAGAAA | CAGCAAGTTCAGCCAAAACA |
| Z21170 | AAACATTGCTTTTGGCCACT | CTCACTCCCCCACACTGTTT |
| kdr_e28 | TATGATAACGCTCCGCCTCT | CAGGGGAATGTCCACAAAAC |

**Table 4.** Genotypes encountered in progeny from intercrosses of *Pdgfra* heterozygotes at E9.5.

| | Total | Wild-type | Heterozygous mutant | Homozygous mutant |
|---|---|---|---|---|
| Number of embryos | 108 | 30[*] | 64[*] | 14[†] |
| Observed ratio | | 0.9 | 2.0 | 0.4[‡] |
| Expected ratio | | 1.0 | 2.0 | 1.0 |

[*] No observed phenotype.

[†] See **Table 5** for detail on observed phenotypes.

[‡] Chi-squared test $p < 0.05$ compared to expected.

cardiomyocytes that reach the midline (**Figure 5A–C**), while *pdgfra* expression appears to be absent from these cells (**Figure 4D**).

Similarly, we found that mouse *Pdgfra* is expressed in the ALPM at E7.5 (**Figure 3F**) and later becomes confined to the caudal aspect of the forming heart tube by E8.0 (**Figure 3G**) and to the inflow tract of the looping heart (**Figure 3H**) as well as the dorsal mesocardium (**Prall et al., 2007**) by E8.5. In more mature hearts, *Pdgfra* is expressed in the atrioventricular valves and epicardium (**Chong et al., 2011**; **Orr-Urtreger et al., 1992**). Even though *Pdgfra* mRNA levels had declined in the anterior cardiac mesoderm by the beginning of heart tube formation (**Figure 3G**), we found persistent PDGFRα protein expression in the forming heart at this stage (**Figure 3I**). PDGFRα was also found in the more caudal domains defined by *Pdgfra* mRNA expression, including the caudal aspect of the forming heart corresponding to its future inflow tract and coelomic mesothelium (**Figure 3I**) (**Bax et al., 2010**).

We did not observe *pdgfra* expression within the anterior endoderm during cardiac fusion in either zebrafish or mouse (**Figures 3G** and **4K–P**; [**Prall et al., 2007**]). In zebrafish, comparison of *axial* and *pdgfra* expression demonstrated mutually exclusive expression domains (**Figure 4K–P**). Lack of *pdgfra* expression in the anterior endoderm is also consistent with previous expression analysis in mouse (**Orr-Urtreger and Lonai, 1992**), as well as with the lack of anterior endoderm defects in *ref* mutant embryos (**Figure 1—figure supplement 2**). Altogether, the *pdgfra* expression patterns in both zebrafish and mouse indicate that *pdgfra* could act within the ALPM to regulate the progression of cardiac fusion.

## *pdgfra* controls the medial direction of cardiomyocyte movement during cardiac fusion

Although our analysis pointed toward a role for *pdgfra* within the ALPM during cardiac fusion, we also considered the possibility that *pdgfra* expression in the early embryo (**Ataliotis et al., 1995**; **Liu et al., 2002**; **Mercola et al., 1990**; **Yang et al., 2008**) could indirectly affect cardiac fusion by influencing processes such as mesoderm involution during gastrulation (**Yang et al., 2008**). However, we did not observe any defects in the size, shape, or bilateral spacing of the ALPM in *ref* mutants at the 8–12 somite stages (**Figure 5A,D,G**), indicating that early ALPM morphogenesis is intact in these embryos. Moreover, we found that pharmacological inhibition of Pdgfr activity at the tailbud stage can disrupt cardiac fusion (**Figure 5—figure supplement 1**; **Figure 2—figure supplement 2**), further

**Table 5.** Cardiac phenotypes observed in *Pdgfra* mutants at E9.5.

| | Number of embryos |
|---|---|
| Total | 14 |
| Normal phenotype | 2 |
| Abnormal looping | 5 |
| Split inflow/common atrial region | 6 |
| Cardia bifida | 1 |

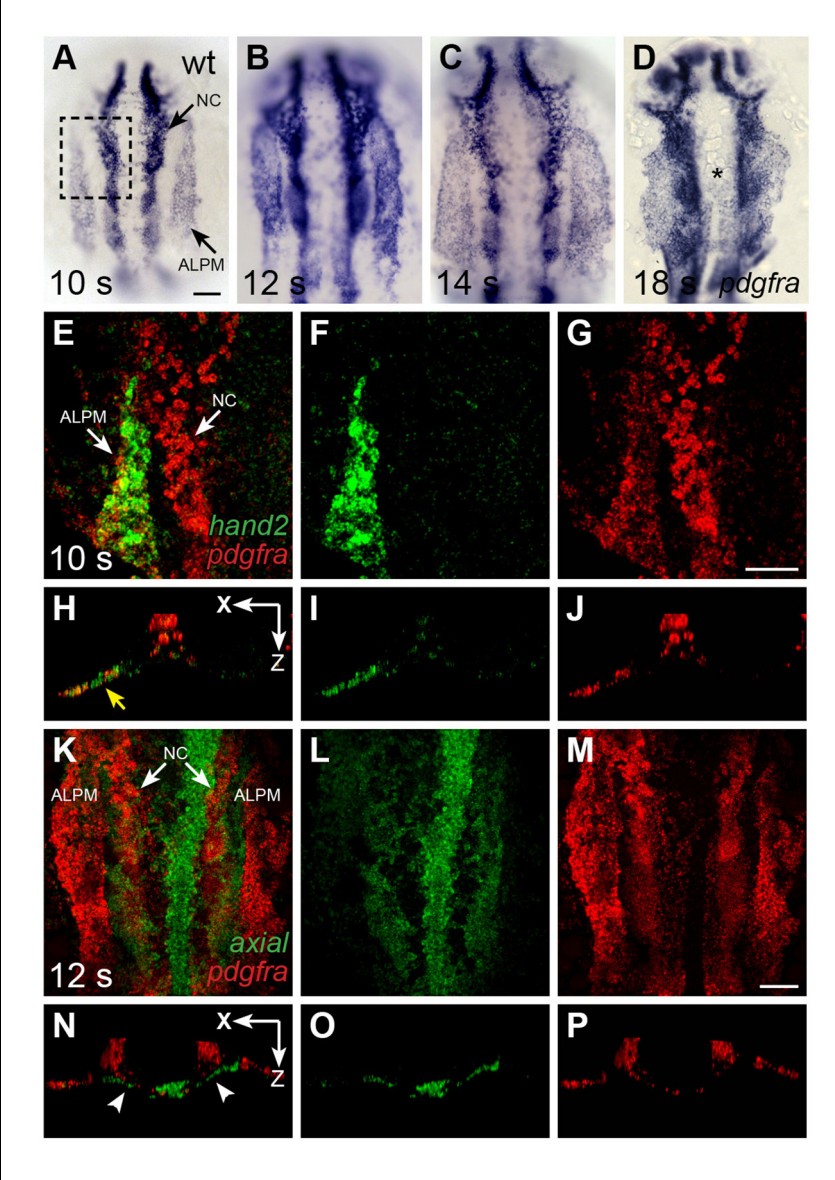

**Figure 4.** *pdgfra* is expressed within the ALPM while cardiac fusion is underway. (A–D) Dorsal views, anterior to the top, depict *pdgfra* expression in wt embryos at 10 s (A), 12 s (B), 14 s (C), and 18 s (D). Arrows (A) indicate *pdgfra* expression in the ALPM and the neural crest (NC). Asterisk (D) denotes position of the myocardium by 18 s; although *pdgfra* is expressed in the myocardial precursors within the ALPM at earlier stages, its expression in these cells appears to be gone by this time point. (E–J) Comparison of *hand2* (green) and *pdgfra* (red) expression patterns demonstrates their overlap in the wt ALPM at 10 s. (E–G) Three-dimensional confocal reconstructions of dorsal views, anterior to the top, focused on the left side of the ALPM (area outlined by a dashed box in (A). Arrows (E) indicate *pdgfra* expression in the ALPM and the NC. (H–J) Single transverse (XZ) sections from (E–G), respectively. Yellow arrow (H) indicates overlap of *hand2* and *pdgfra* expression in the ALPM. (K–P) Comparison of *axial* (green) and *pdgfra* (red) expression patterns demonstrates lack of *pdgfra* expression in the *axial*-expressing anterior endoderm in wt embryos at 12 s. (K–M) Three-dimensional confocal reconstructions of dorsal views, anterior to the top; arrows (K) indicate *pdgfra* expression in the ALPM and the neural crest. (N–P) Single transverse (XZ) sections from (K–M), respectively. Arrowheads (N) indicate *axial* expression in the anterior endoderm, adjacent to, but not overlapping with, *pdgfra* expression in the ALPM. Scale bars: 60 μm.

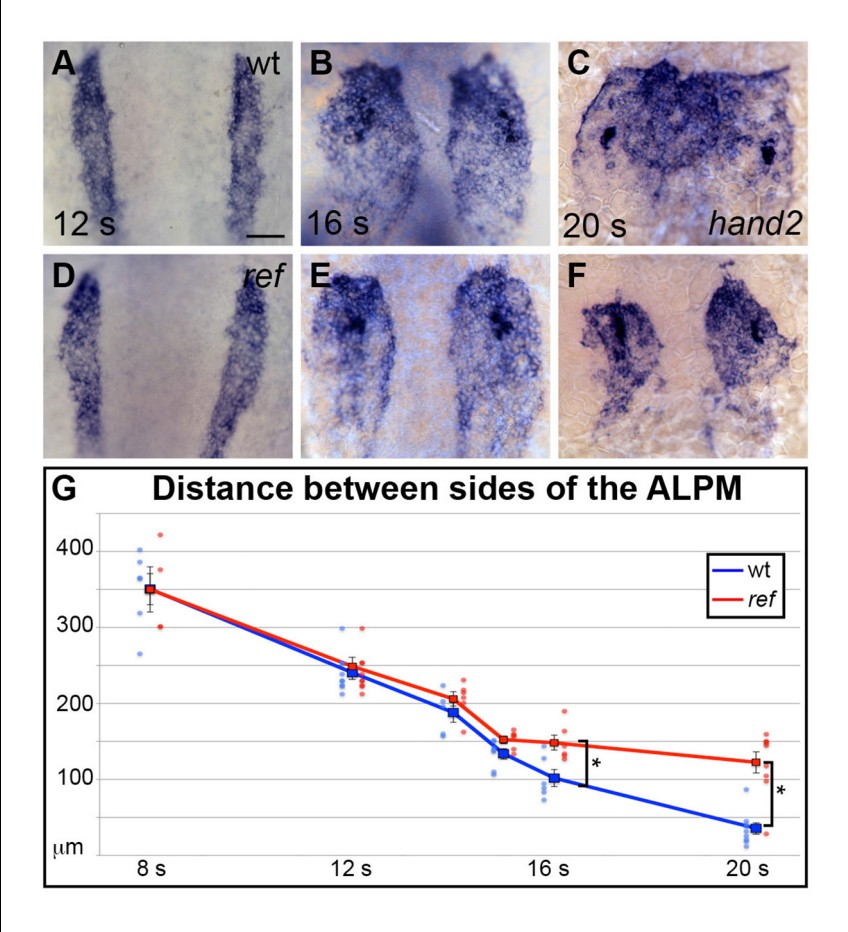

**Figure 5.** *pdgfra* influences the movement of the ALPM after the 15 somite stage. (**A–F**) Dorsal views, anterior to the top, depict expression of *hand2* in the wt (**A–C**) and *ref* mutant (**D–F**) ALPM from 12 to 20 s. The morphology and position of the ALPM are indistinguishable in wt (**A**) and *ref* mutant (**D**) embryos at 12 s. After 15 s (**B,C,E,F**), disrupted movement of the ALPM is evident in *ref* mutants. Scale bar: 60 μm. (**G**) Graph illustrates the average distance between the two sides of the ALPM in wt and *ref* mutant embryos from 8 to 20 s. In each embryo, the distance between the sides of the ALPM was calculated by measuring the distance between the medial edges of the *hand2*-expressing domains at three equidistant points (200 μm apart) along the anterior-posterior axis. The largest of those three measurements was selected as representative of the maximum distance between the bilateral ALPM domains for that embryo. Dots represent the selected measurements from individual embryos. The distance between the bilateral sheets in *ref* mutant embryos begins to diverge significantly from wt after 15 s. Error bars represent the standard error. Asterisks indicate $p < 0.05$ (Student's t-test): $p = 0.99$ at 8 s; $p = 0.58$ at 12 s; $p = 0.30$ at 14 s; $p = 0.053$ at 15 s; $p = 0.012$ at 16 s; and $p = 0.00012$ at 20 s.

The following figure supplements are available for figure 5:

**Figure supplement 1.** PDGF signaling is required after gastrulation for proper cardiac fusion.

**Figure supplement 2.** Overexpression of *pdgfaa* influences ALPM movement by the 15 somite stage.

supporting the conclusion that *pdgfra* activity influences cardiac fusion after gastrulation is complete.

To determine when cardiac fusion first goes awry in *ref* mutants, we began by comparing the distance between the left and right sides of the ALPM in wild-type and *ref* mutant embryos. Until the 15 somite stage, the spacing between the bilateral domains of the ALPM was normal in *ref* mutants (*Figure 5G*). After the 15 somite stage, the *ref* mutants began to display an evident phenotype: whereas the two sides of the wild-type ALPM continued to move toward each other, the sides of the

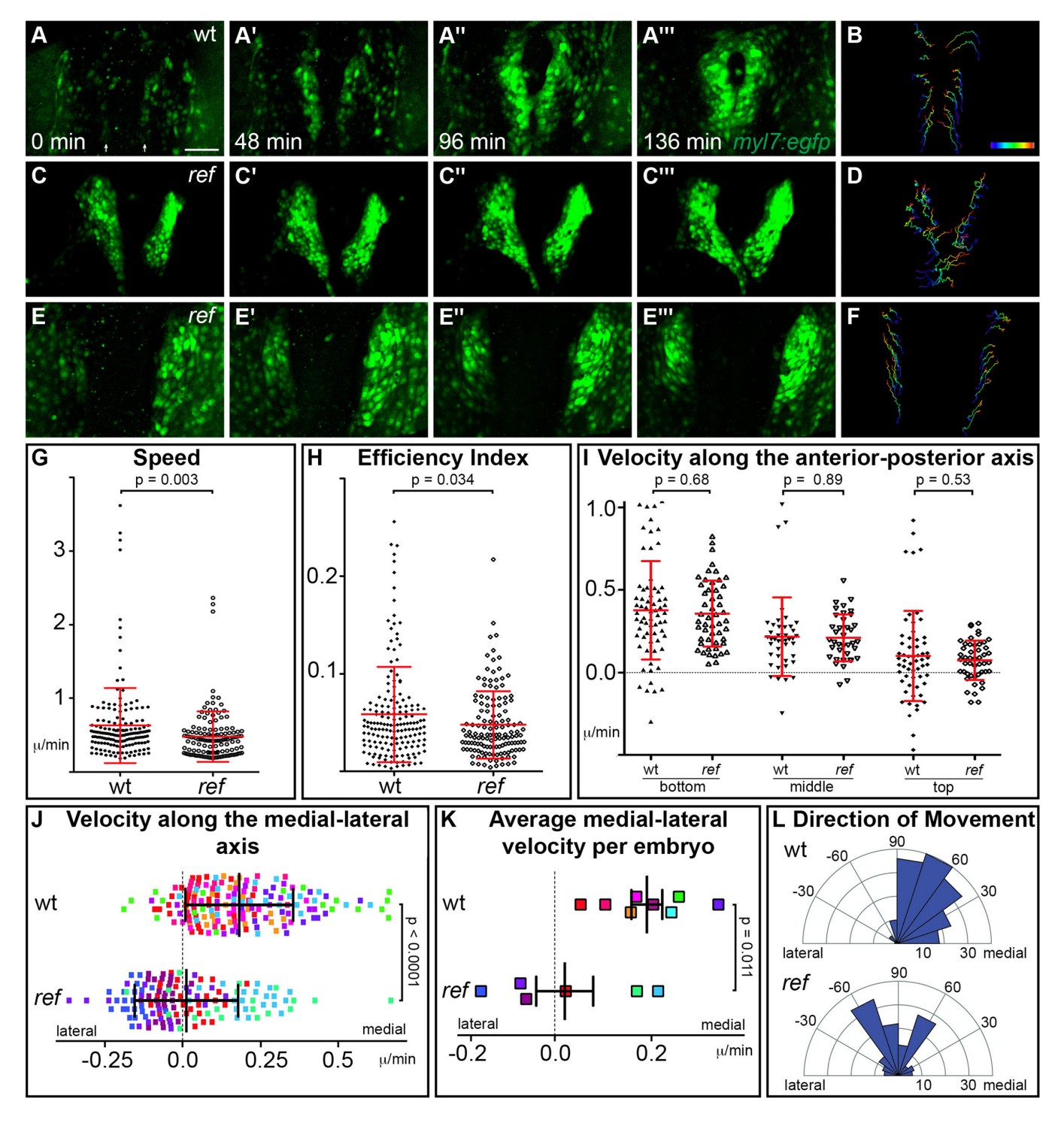

**Figure 6.** *pdgfra* regulates the directionality of cardiomyocyte movement. (A–F) Representative timelapse experiments indicate patterns of cell movement in wt (A,B) and *ref* mutant (C–F) embryos carrying the *Tg(myl7:egfp)* transgene. (A,C,E) Three-dimensional confocal reconstructions of select timepoints within each timelapse depict the typical changes in myocardial morphology seen over time in wt (A), mildly affected *ref* mutant (C) and severely affected *ref* mutant (E) embryos. (B,D,F) Tracks show the movements of the innermost cardiomyocytes in these embryos over the course of a ~2 hr timelapse. Cell tracks are colored from blue to red, indicating the location of each cell from the beginning to the end of its trajectory. See also *Videos 1–3*. Scale bar: 60 µm. (G–L) Quantitative analysis of cardiomyocyte movements. 168 and 137 cells were analyzed from eight wt and six *ref* mutant embryos, respectively. Graphs depict the average speed of individual cells (G, distance/time), the average efficiency index of individual cells (H,

*Figure 6 continued on next page*

*Figure 6 continued*
displacement/distance), the average velocity (displacement/time) of individual cells along the anterior-posterior axis (I) and along the medial-lateral axis (J), the average medial-lateral velocity per embryo (K), and the direction of the overall trajectory of individual cells (L). Dots in (J) are colored to depict the embryo to which they belong, and the same color scheme is used in (K). In (L), individual cells are grouped into 10 bins based on their net direction of movement; length of each radial bar represents the number of cells in each bin. The velocity along the medial-lateral axis (J,K) and the direction of cell trajectories (L) were significantly altered in *ref* mutants, indicating the misdirection of *ref* mutant cardiomyocytes and implicating *pdgfra* in steering the medial direction of cardiomyocyte movement. Error bars represent the standard deviation; p values were determined using Student's T-test.

*ref* mutant ALPM stayed apart (*Figure 5B,C,E,F,G*). Thus, although the initial stages of ALPM convergence are unaffected in *ref* mutants, the *ref* mutant ALPM is unable to approach the midline normally during cardiac fusion.

We next sought to elucidate the cellular defects responsible for the inhibition of cardiac fusion in *ref* mutants. Do *ref* mutant cardiomyocytes move at a sluggish rate or are they misdirected? Previous studies have shown that VEGF signaling can regulate the speed of endocardial precursor movement during cardiac fusion (*Fish et al., 2011*), suggesting the possibility that PDGF signaling might set the pace of myocardial precursor movement. Alternatively, PDGF signaling has been shown to control the direction of mesodermal movement during gastrulation (*Damm and Winklbauer, 2011*; *Nagel et al., 2004*), suggesting that it could also guide the route taken by myocardial cells during cardiac fusion. To test these hypotheses, we tracked individual cell movements over time, using the myocardial reporter transgene *Tg(myl7:egfp)* (*Holtzman et al., 2007*; *Huang et al., 2003*) to follow the patterns of cardiomyocyte behavior in live embryos (*Figure 6*).

We initiated our timelapse analysis at the 16 somite stage, the earliest timepoint when we could robustly detect *Tg(myl7:egfp)* expression. Consistent with our analysis of ALPM position (*Figure 5*), the bilateral populations of cardiomyocytes in *ref* mutants were already slightly farther apart than their wild-type counterparts were at the 16 somite stage (*Figure 6A,C,E*). By following the movements of these cells during cardiac fusion, we found that wild-type cardiomyocytes display a coherent pattern of collective movement without significant neighbor exchange (*Figure 6A,B*), consistent with our prior work (*Holtzman et al., 2007*). Cardiomyocytes in *ref* mutants exhibited similar patterns of coherent movement (*Figure 6C–F*). However, while wild-type cardiomyocytes moved progressively toward the midline (*Figure 6A,B*; *Video 1*), the medial movement of *ref* mutant cardiomyocytes seemed severely diminished, even though these cells still appeared to be in motion (*Figure 6C–F*; *Videos 2–3*). In *ref* mutants with a relatively mild phenotype, a posterior subset of cardiomyocytes still exhibited sufficient medial movement to fuse at the midline (*Figure 6C,D*; *Video 2*). In more severely affected *ref* mutants, medial movement appeared lost along the entire anterior-posterior extent of the myocardium (*Figure 6E,F*; *Video 3*).

Lack of medial movement could be the result of defects in several aspects of cell behavior including speed, efficiency, and directionality. To distinguish between these possibilities, we performed quantitative analysis of individual cardiomyocyte trajectories. Compared to wild-type cardiomyocytes, *ref* mutant cardiomyocytes moved at a slightly slower average speed

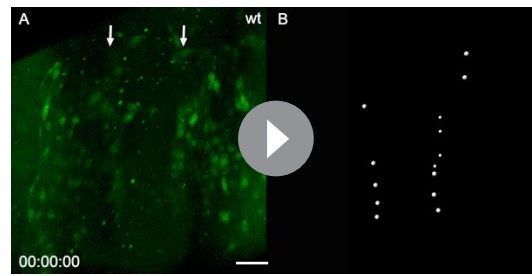

**Video 1.** Cardiomyocytes in a wild-type embryo undergo medially directed movement during cardiac fusion. (A,B) Representative timelapse movie (A) and associated tracks (B) of cardiomyocyte movement occurring during cardiac fusion in a wild-type embryo carrying the *Tg(myl7:egfp)* transgene. (A) Movie of drift-corrected three-dimensional reconstructions of 30 confocal slices taken at ~4 min intervals for ~2 hr, starting when eGFP could first be detected in the ALPM. (B) The movements of individual cardiomyocytes at the innermost region of the ALPM were tracked (dots, B) at each time point. Their positions over the previous 80 min are depicted as connected colored tracks (blue-to-red, beginning-to-end). Blank frames indicate brief pauses in acquisition for refocusing. Arrows indicate initial starting position of cardiomyocytes. Asterisks indicate GFP[+] cells that are not cardiomyocytes. Scale bar: 40 μm.

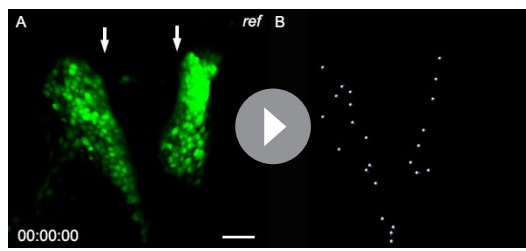

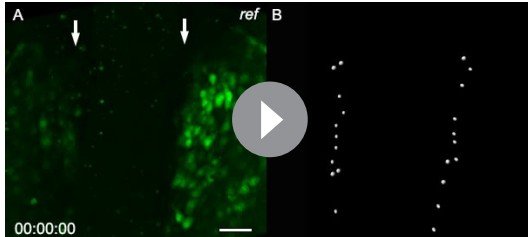

**Video 2.** Not all cardiomyocytes in a mildly affected *ref* mutant embryo undergo medially directed movement during cardiac fusion. (A,B) Representative timelapse movie (A) and associated tracks (B) of cardiomyocyte movement in a mildly affected *ref* mutant embryo carrying the *Tg(myl7:egfp)* transgene. Images were acquired as described for *Video 1*; however, drift correction was not applied to this movie and thus its tracks were not included in further quantitative analysis. In mildly affected *ref* mutant embryos, posterior cardiomyocytes display sufficient medial movement to fuse at the midline, while anterior cardiomyocytes do not. Arrows indicate initial starting position of cardiomyocytes. Asterisks indicate GFP[+] cells that are not cardiomyocytes. Scale bar: 40 μm.

**Video 3.** Cardiomyocytes in a severely affected *ref* mutant embryo fail to display medially directed movement during cardiac fusion. (A,B) Representative timelapse movie (A) and associated tracks (B) of cardiomyocyte movement in a severely affected *ref* mutant embryo carrying the *Tg(myl7:egfp)* transgene. Images were acquired as described for *Video 1*, with drift correction. In severely affected *ref* mutant embryos, none of the cardiomyocytes display measurable medial movement. Arrows indicate initial starting position of cardiomyocytes. Blank frames indicate brief pauses in acquisition for refocusing. Asterisks indicate GFP[+] cells that are not cardiomyocytes. Scale bar: 40 μm.

(distance/time) (*Figure 6G*) and with a slightly reduced efficiency (displacement/distance) (*Figure 6H*). When examining velocity (displacement/time) along particular axes, we found no difference between the velocities of wild-type and *ref* mutant cardiomyocyte movement along the anterior-posterior axis (*Figure 6I*). However, there was a substantial difference between the velocities of wild-type and *ref* mutant cardiomyocyte movement along the medial-lateral axis: the average velocity along the medial-lateral axis was 0.19 micron/min for wild-type cardiomyocytes, but was only 0.016 micron/min for *ref* mutant cardiomyocytes (*Figure 6J*). This difference in cell behavior becomes even more striking when considering the variability in the *ref* mutant phenotype. Two of the six *ref* mutant embryos examined had a relatively mild phenotype, and the cardiomyocytes in these embryos exhibited an average medial-lateral velocity similar to that seen in wild-type embryos (*Figure 6K*). In contrast, the other four *ref* mutant embryos displayed a more severe phenotype, and the cardiomyocytes in these embryos had an average medial-lateral velocity near or below zero (*Figure 6K*). Further examination of the vectors of cell movement revealed that these deficiencies in medial-lateral velocity reflect the misdirection of *ref* mutant cardiomyocytes. In our wild-type timelapse data, almost all cardiomyocytes move in the medial direction, whereas over half of the cardiomyocytes in our *ref* mutant timelapse data show no medial movement, with many of these cells moving away from the midline (*Figure 6L*). Together, these data reveal that *pdgfra* plays an important role in steering cardiomyocyte movement toward the midline during cardiac fusion.

## The Pdgfra ligand *pdgfaa* is expressed in the anterior endoderm, adjacent to the ALPM

We next evaluated whether the expression patterns of genes encoding Pdgfra ligands could provide insight into how PDGF signaling confers directionality to cardiomyocyte movement. Initial examination demonstrated that both *pdgfaa* and *pdgfab*, but not *pdgfc*, are expressed in bilateral medial stripes within the anterior portion of the embryo (*Figure 7—figure supplement 1*). Deeper analysis of *pdgfaa* expression revealed that it is expressed in bilateral domains within the anterior endoderm between the 10 and 16 somite stages, positioned near the lateral edges of this tissue (*Figure 7A–J*). The expression of *pdgfaa* within the anterior endoderm in zebrafish is grossly consistent with prior studies demonstrating expression of *Pdgfa* in the mouse foregut (*Palmieri et al., 1992*). We readdressed this issue in mouse and found *Pdgfa* expression in endoderm at the rim of the foregut pocket (*Figure 3J*) as well as in the pharyngeal floor and pharyngeal pouches (*Figure 3J'*), regions

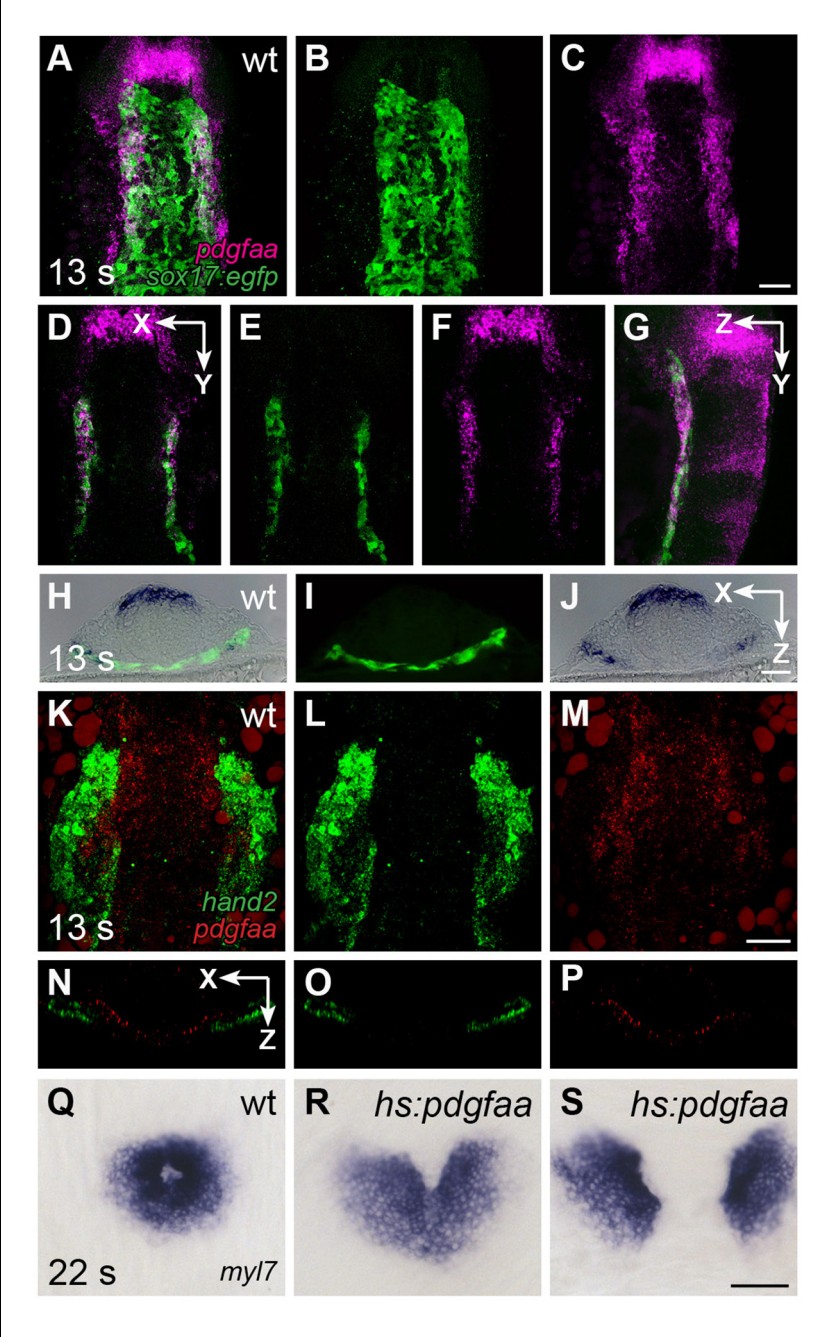

**Figure 7.** *pdgfaa* is expressed in the anterior endoderm, medially adjacent to the ALPM. (**A–G**) Fluorescent in situ hybridization and immunofluorescence compare the expression of *pdgfaa* (magenta) and *Tg(sox17:egfp)* (green) in wt embryos at 13 s. (**A–C**) Dorsal views, anterior to the top, of a three-dimensional reconstruction show that *pdgfaa* is expressed in bilateral domains of the anterior endoderm, near the lateral edges of the endodermal sheet. Expression of *pdgfaa* is also evident in cranial rhombomeres. (**D–F**) A coronal (XY) slice of the same embryo demonstrates the overlap between *pdgfaa* and *Tg(sox17:egfp)* expression. (**G**) A sagittal (ZY) slice of the same embryo provides a lateral view. (**H–J**) Transverse cryosections compare the expression of *pdgfaa* (blue) and the expression of *Tg(sox17:egfp)* (green) in wt embryos at 13 s, showing that *pdgfaa* is expressed in lateral domains of the endodermal sheet. (**K–P**) Comparison of *hand2* (green) and *pdgfaa* (red) expression patterns demonstrates that *pdgfaa* is expressed medially adjacent to the domains of *hand2* expression in the ALPM. (**K–M**) Three-dimensional confocal reconstructions of dorsal views, anterior to the top. (**N–P**) Single transverse (XZ) slices from (**K–M**), respectively. (**Q–S**) Dorsal views, anterior to the top, display the expression of *myl7* at 22 s in nontransgenic (**Q**) or *Tg(hsp70l:pdgfaa-2A-mCherry)* (**R,S**) embryos, following heat shock at the tailbud stage. Ectopic expression

*Figure 7 continued on next page*

*Figure 7 continued*

of *pdgfaa* causes cardiac fusion defects. See *Figure 2—figure supplement 2* and *Figure 5—figure supplement 2* for additional information regarding this phenotype. Scale bars: 60 µm.

The following figure supplement is available for figure 7:

**Figure supplement 1.** Expression of genes encoding Pdgfra ligands.

closely associated with the forming heart tube at E8.0 and earlier stages. Moreover, the *pdgfaa* expression domains in zebrafish are medially adjacent to the positions of the myocardial precursors within the ALPM (*Figure 7K–P*), suggesting the possibility of a paracrine relationship between Pdgfa ligands in the endoderm and Pdgfra in the ALPM.

To investigate whether the spatially restricted expression of *pdgfa* genes is important for the regulation of cardiac fusion, we induced *pdgfaa* expression throughout the embryo using the heat-inducible transgene *Tg(hsp70l:pdgfaa-2A-mCherry)*. Following heat shock at the tailbud stage, transgenic embryos displayed cardiac fusion defects similar to those observed in *ref* mutants (*Figure 7Q–S*; *Figure 2—figure supplement 2*; *Figure 5—figure supplement 2*). The ability of ectopic *pdgfaa* expression to disrupt cardiac fusion indicates that PDGF signaling has the potential to serve as an instructive influence in directing cardiomyocytes toward the midline.

## Discussion

Taken together, our studies point to a model in which the PDGF signaling pathway facilitates communication between the endoderm and the myocardium and thereby directs cardiomyocytes toward the midline during cardiac fusion. We propose that Pdgfa ligands, produced by the anterior endoderm, signal through the Pdgfra receptor in the ALPM in order to control the directionality of cardiomyocyte behavior. This connection parallels other examples in which PDGF ligand-receptor pairs in adjacent tissues influence cell movements (e.g. in the kidney, neural crest, and gastrulating mesoderm [*Eberhart et al., 2008*; *Lindahl et al., 1998*; *Yang et al., 2008*]), highlighting a paradigm for how tissue-tissue interactions establish the landscape of organogenesis (*Andrae et al., 2008*; *Hoch and Soriano, 2003*; *Orr-Urtreger and Lonai, 1992*). Importantly, our demonstration of this new function for PDGF signaling represents the first insight into a pathway that guides the medial direction of cardiomyocyte movement. Moreover, our findings suggest a previously unappreciated molecular basis for the interactions between the endoderm and the myocardium that govern cardiac fusion. Further elucidation of the paracrine relationship between Pdgfa ligands in the endoderm and the Pdgfra receptor in the ALPM awaits the development of appropriate tools for the tissue-specific inactivation of these players.

How might PDGF signaling confer directionality on the collective behavior of the cardiomyocytes? In the absence of *pdgfra* function, myocardial movement is no longer directed toward the midline, implicating PDGF signaling in the arrangement of the forces that steer this epithelial tissue. One intriguing possibility is that PDGF signaling could direct medial movement via polarized Pdgfra activation that controls oriented formation of active protrusions, akin to the activity of the PDGF/VEGF receptor Pvr in *Drosophila*, which directs the collective movement of the epithelial border cells during oogenesis and the epidermal cells during dorsal closure (*Duchek et al., 2001*; *Garlena et al., 2015*). Alternatively, PDGF signaling could promote other types of epithelial reorganization that could facilitate directional movement, such as the rearrangement of adherens junctions or extracellular matrix at the medial edge of the myocardium, causing epithelial deformations that drive movement forward (*Xu et al., 2005*; *Yang et al., 2008*). Future examination of the relationship of PDGF signaling to the subcellular characteristics of the myocardium during cardiac fusion will help to elucidate the precise morphogenetic consequences of Pdgfra activity.

It is likely that the PDGF signaling pathway works in parallel with additional pathways to influence cardiac cell movement during cardiac fusion. Although *ref* mutants fail to undergo proper cardiac fusion, they do not fully phenocopy mutants with primary endoderm defects (e.g. *casanova* (*cas*; *sox32*) or *miles apart* (*mil*; *s1pr2*) [*Alexander et al., 1999*; *Kikuchi et al., 2001*; *Kupperman et al., 2000*; *Ye and Lin, 2013*; *Yelon et al., 1999*]). In *cas* and *mil* mutants, both the endocardial and

myocardial precursors fail to move to the midline (*Holtzman et al., 2007*; *Wong et al., 2012*; *Xie et al., 2016*); moreover, their myocardial movement defects can be detected prior to the eight somite stage (*Ye et al., 2015*). In contrast, the endocardial precursors seem to reach the midline normally in *ref* mutants, and the *ref* myocardial movement defects emerge only after the 15 somite stage. Most likely, other factors, such as VEGF signaling to the endocardium (*Fish et al., 2011*) or mechanical forces from the endoderm (*Aleksandrova et al., 2015*; *Varner and Taber, 2012*), collaborate with PDGF signaling to control distinct aspects of endocardial and myocardial cell behavior during earlier and later phases of cardiac fusion.

The cardiomyocyte movements that occur during cardiac fusion, guided by interactions with the endoderm, establish a foundation of proper tissue orientation and morphology upon which to assemble the initial heart tube. Our studies in zebrafish and mouse reveal a conserved influence of PDGF signaling on heart tube assembly. This influence is likely to be relevant to CHD in humans, since defects in cardiac morphology can originate in the cardiac precursor populations involved in cardiac fusion and heart tube assembly (*Prall et al., 2007*; *Vincent and Buckingham, 2010*). More generally, the molecular mechanisms that control the direction of cardiomyocyte movement are likely to be relevant to the etiology of disorders that are caused by anomalous cell movement, potentially including ventricular septal defects, atrial septal defects, outflow tract defects, and trabeculation abnormalities (*Bax et al., 2010*; *Bruneau, 2008*; *Ding et al., 2004*; *Neeb et al., 2013*; *Samsa et al., 2013*), as well as inflow tract defects known to be associated with mutations in *PDGFRA* (*Bleyl et al., 2010*). *PDGFRA* is deleted in a number of human families showing total anomalous pulmonary venous return (TAPVR), in which the pulmonary arteries connect with the systemic venous system instead of the left atrium, a defect replicated in mouse and chick loss-of-function models (*Bleyl et al., 2010*). TAPVR occurs in 1 in 15,000 live births and, while life-threatening, is at the mild end of the spectrum of morphogenetic defects that we have observed in *Pdgfra* knockout mice. Thus, our studies suggest the possibility of a broader involvement of *PDGFRA* mutations in CHD, specifically through their effects on heart tube assembly, and more globally as part of the spectrum of diseases associated with aberrant cardiac cell movements.

## Materials and methods

### Zebrafish

We used the following transgenic and mutant strains of zebrafish: *Tg(myl7:egfp)^twu34* (*Huang et al., 2003*) (RRID:ZFIN_ZDB-GENO-050809-10), *Tg(fli1a:negfp)^y7* (*Roman et al., 2002*) (RRID:ZFIN_ZDB-GENO-060821-2), *Tg(sox17:egfp)^ha01* (*Mizoguchi et al., 2008*) (RRID:ZFIN_ZDB-GENO-080714-2), pdgfra^b1059 (*Eberhart et al., 2008*) (RRID:ZFIN_ZDB-GENO-081008-1), and *ref* (*pdgfra^sk16*; this paper). The *Tg(hsp70l:pdgfaa-2A-mCherry)^sd44* transgene was assembled using the D-Topo vector (Invitrogen, Carlsbad, CA) with a *pdgfaa* cDNA lacking the stop codon (*Eberhart et al., 2008*), in combination with established Gateway cloning vectors (*Kwan et al., 2007*). The final destination vector was created by inserting a Cryaa:CFP cassette (*Hesselson et al., 2009*) into the pDestTol2pA4 vector (gift from K. Kwan). Transgenic founders were established using standard techniques for Tol2-mediated transgenesis (*Fisher et al., 2006*). We analyzed F2 embryos from four separate transgenic lines to evaluate the effect of *pdgfaa* overexpression on cardiac fusion. Embryos were heat shocked at 38°C for 45 min beginning at the tailbud stage and were then returned to 28°C. Transgenic and nontransgenic sibling embryos were distinguished based on their expression of mCherry following heat shock. All zebrafish work followed protocols approved by the UCSD IACUC.

### Mice

*Pdgfra* null embryos were generated by intercrossing heterozygous *Pdgfra^tm11(EGFP)Sor* (*Hamilton et al., 2003*) (RRID:MGI:5519063) mutant mice on a co-isogenic C57BL/6J background. In situ hybridization and immunohistochemistry were performed using standard protocols (*Prall et al., 2007*), and genotyping was performed as described for Jax stock #007669 (https://www.jax.org/strain/007669). Images were captured using a Leica M125 microscope outfitted with a Leica DFC295 camera and processed using Adobe Photoshop. All mouse experiments were overseen by the Garvan Institute of Medical Research/St. Vincent's Hospital Animal Ethics Committee.

## Positional cloning and genotyping

Meiotic recombinants were mapped using polymorphic SSLP and SNP markers to identify a small critical interval on linkage group 20; PCR primers used for mapping are provided in *Table 3*. Sequence analysis of candidate genes was performed on cDNA from homozygous wild-type and *ref* mutant embryos.

PCR genotyping was used to identify *ref* mutant embryos following phenotypic analysis. The primer pair 5'-GTAGGTAAAAGTAAAGCTGGTA-3' and 5'-CAAGGGTGTGTTGAACCTGA-3' amplifies a 136 bp PCR product flanking the e14i15 boundary in the *pdgfra* locus and creates a KpnI restriction site within the wild-type allele, but not within the *ref* mutant allele. Digestion of the wild-type PCR product with KpnI creates fragments of 113 and 23 bp.

## Morpholinos and inhibitors

A *pdgfra* morpholino (5'-CACTCGCAAATCAGACCCTCCTGAT-3') was designed to disrupt the splicing of exon 11 and thereby lead to premature truncation of Pdgfra prior to its kinase domain. We injected 12 ng of morpholino at the one-cell stage; this dose did not induce visible toxicity. Furthermore, injection of this morpholino into *ref* mutants did not increase the frequency or severity of their cardiac fusion defects.

For pharmacological inhibition of PDGF signaling (*Kim et al., 2010*), we incubated embryos in Pdgfr inhibitor V (Calbiochem 521234, Temecula, CA) from the tailbud stage until the 22 somite stage. Three separate experiments were performed, using doses of 0.25–0.4 µM.

## In situ hybridization, immunofluorescence, and Alcian blue staining

The following probes and antibodies were used: *myl7* (ZDB-GENE-991019–3), *axial/foxa2* (ZDB-GENE-980526–404, *sox17* (ZDB-GENE-991213–1), *hand2* (ZDB-GENE-000511–1), *pdgfra* (ZDB-GENE-990415–208), *pdgfaa* (ZDB-GENE-030918–2), *pdgfab* (ZDB-GENE-060929–124), *pdgfc* (ZDB-GENE-071217–2), anti-GFP (RRID:AB_300798; Abcam ab13970; 1:1000), anti-ZO-1 (RRID:AB_2533147; Zymed 33–9100; 1:200), and donkey anti-mouse Alexa 488 (RRID:AB_141607; Invitrogen; 1:300). Standard in situ hybridization, fluorescent in situ hybridization, and immunofluorescence were performed using established protocols (*Alexander et al., 1998*; *Brend and Holley, 2009*; *Yelon et al., 1999*). Fluorescent in situ hybridization was combined with immunofluorescence as previously described (*Zeng and Yelon, 2014*). Standard in situ hybridization was combined with visualization of transgene expression by creating transverse sections following in situ hybridization, using standard cryoprotection, embedding, and sectioning techniques (*Garavito-Aguilar et al., 2010*) and then performing standard immunofluorescence for GFP on sections. Alcian blue staining was performed as previously described (*Kimmel et al., 1998*). Trunks were removed for genotyping prior to Alcian staining.

Images were captured using Zeiss M2Bio, AxioZoom and AxioImager microscopes outfitted with Axiocam cameras and processed with Adobe Photoshop. Confocal stacks were collected using a Leica SP5 confocal laser-scanning microscope and processed using Imaris (Bitplane, Belfast, Ireland).

## Timelapse imaging and cell tracking

*Tg(myl7:egfp)* embryos at the 14 somite stage were mounted head down in 0.8% low-melt agarose and placed on a coverslip bottom dish in wells made from a layer of 3% agarose. Timelapse images were collected using a Leica SP5 confocal microscope with a 20X objective, in a chamber heated to 28°C. Confocal stacks of GFP and brightfield images were collected every 4 min for 2–3 hr, starting around the 16 somite stage. In each stack, 30 confocal slices spanning the expression of *Tg(myl7: egfp)* were collected at ~3 µm intervals. Embryos were retained after completion of imaging, and we only analyzed data from embryos that appeared healthy for 24 hr following the timelapse.

Image processing and cell tracking was performed on three-dimensional reconstructions generated with Imaris, using the semi-automated cell tracking module. In each embryo, we tracked 20–30 cells from the two most medial columns of cardiomyocytes on each side. Only tracks in which a cell position could be determined for each timepoint were used for further analysis. We also tracked the tip of the notochord in brightfield images at each timepoint. Although we observed a slight posterior retraction of the notochord over the course of our timelapse analysis, we found that this was the most consistent landmark to use as a reference point to correct for drift that occurred during

imaging. Thus, the movement of the tracked notochord tip was subtracted from the movement of each tracked cardiomyocyte. Our wild-type tracking data were largely consistent with our prior studies (*Holtzman et al., 2007*), including the velocity of movement, coherence of movement, lack of cell movement in the Z-axis, and direction of wild-type cardiomyocyte trajectories. Subtle differences between these two data sets are likely due to our current use of the notochord as a reference point and the slightly later stage at which we initiated these timelapse experiments.

For quantitative analysis of cardiomyocyte movement, we extracted the X and Y position of each cell at each timepoint along its track, as previously described (*Holtzman et al., 2007*). Cell movement properties, including overall speed (distance/time), efficiency (displacement/distance), velocity (displacement/time), and direction, were then calculated for each individual cardiomyocyte. Velocity measurements were split into their X (medial-lateral) and Y (anterior-posterior) components. Cells along the anterior-posterior axis were further divided into top, middle, and bottom subsets, as in our prior work (*Holtzman et al., 2007*). Direction was calculated as arctan[abs(y-displacement)/(x-displacement)], after aligning movement between the left and right sides. Graphs were made using Matlab (Mathworks, Natick, MA) and Prism (Graphpad, La Jolla, CA) software.

## Statistics and replication

All statistical analyses were performed using a two-tailed unpaired Student's t-test. No statistical methods were used to predetermine sample sizes. Instead, sample sizes were determined based on prior experience with relevant phenotypes and standards within the zebrafish and mouse communities. All results were collected from at least two independent experiments (technical replicates) in which multiple embryos, from multiple independent matings, were analyzed (biological replicates).

## Acknowledgements

We thank members of the Yelon lab and N Chi, S Evans, D Traver, P Soriano, G Crump, and J Schoenebeck for valuable discussions, as well as J Eberhart and K Kwan for providing reagents.

## Additional information

### Competing interests

RPH: Reviewing editor, *eLife*. The other authors declare that no competing interests exist.

### Funding

| Funder | Grant reference number | Author |
|---|---|---|
| American Heart Association | 12POST11660038 | Joshua Bloomekatz |
| Australian Heart Foundation | CR 08S 3958 | Owen WJ Prall |
| National Health and Medical Research Council | 573732 | Owen WJ Prall<br>Richard P Harvey |
| National Health and Medical Research Council | 573707 | Owen WJ Prall |
| National Health and Medical Research Council | 1074386 | Richard P Harvey |
| National Health and Medical Research Council | 573705 | Richard P Harvey |
| Australian Research Council | Stem Cells Australia SR110001002 | Richard P Harvey |
| National Heart, Lung, and Blood Institute | R01HL081911 | Deborah Yelon |
| March of Dimes Foundation | 1-FY11-493 | Deborah Yelon |
| National Heart, Lung, and Blood Institute | R01HL133166 | Deborah Yelon |

The funders had no role in study design, data collection and interpretation, or the decision to submit the work for publication.

## Author contributions

JB, Conceptualization, Formal analysis, Funding acquisition, Investigation, Writing—original draft, Writing—review and editing; RS, Conceptualization, Formal analysis, Investigation, Writing—original draft, Writing—review and editing; OWJP, Conceptualization, Formal analysis, Funding acquisition, Investigation, Writing—review and editing; ACD, MV, C-SL, Investigation, Writing—review and editing; RPH, DY, Conceptualization, Formal analysis, Funding acquisition, Writing—original draft, Project administration, Writing—review and editing

## Author ORCIDs

Joshua Bloomekatz, http://orcid.org/0000-0001-5816-2756
Reena Singh, http://orcid.org/0000-0001-8860-8800
Owen WJ Prall, http://orcid.org/0000-0003-1600-5933
Ariel C Dunn, http://orcid.org/0000-0002-1715-3682
Megan Vaughan, http://orcid.org/0000-0002-8289-0638
Chin-San Loo, http://orcid.org/0000-0001-7915-2995
Richard P Harvey, http://orcid.org/0000-0002-9950-9792
Deborah Yelon, http://orcid.org/0000-0003-3523-4053

## Ethics

Animal experimentation: All zebrafish work followed protocols approved by the UCSD IACUC (protocol S09125). All mouse experiments were overseen by the Garvan Institute of Medical Research/St. Vincent's Hospital Animal Ethics Committee (projects AEC13/01 and AEC13/02).

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
