## [Decision Letter]

Thank you for submitting your article "PDGF signaling directs cardiomyocyte movement toward the midline during heart tube assembly" for consideration by *eLife*. Your article has been favorably evaluated by Marianne Bronner (Senior Editor) and three reviewers, one of whom is a member of our Board of Reviewing Editors. The reviewers have opted to remain anonymous.

The reviewers have discussed the reviews with one another and the Reviewing Editor has drafted this decision to help you prepare a revised submission.

Summary:

This paper from the Yelon lab sheds critical light on early cardiac morphogenesis, namely the movement of the precardiac mesoderm towards the midline. Although it was shown many years ago that the endoderm is important for this process, it has been mostly unclear how it is involved. Starting with a forward genetic screen, the authors identify the role of Pdgf signaling in the process, with the precardiac mesoderm expressing a Pdgf receptor and the endoderm expressing a Pdgf ligand. Further experiments indicate that the Pdgf signaling could be instructive in the movement of the precardiac mesoderm towards the midline. Additional data indicate that Pdgf signaling is also involved in early mouse cardiac development. The paper is very well written and the data of high quality.

Essential revisions:

1) The concept of paracrine signaling should be further tested by overexpressing *pdgfaa* specifically in the endoderm. In addition, *pdgfc* (another potential ligand for Pdgfra) expression should be analyzed at the relevant stages.

[The endodermal-myocardial connection should be further developed, also as it is the title of the manuscript.]

2) The penetrance of the phenotypes is hard to gauge for both the fish and mouse data. This would be best represented in a table, instead of the figure legends (Figure 1—figure supplement 1 and Figure 3), where some (but not all) of the data is present. See below:

Fish: The authors must have genotyped all of the embryos, how many of these are "morphologically evident mutants"? It is important to provide this information as this may significantly reduce the incidence of the mutants.

Fish, if the authors only see four cardia bifida mutants, and the incidence is also low in the mouse, mention of this phenotype should be removed from the abstract as this is misleading (not the dominant phenotype).

Mouse: can some of the defects in heart development simply be a delay? In the old Grüneberg papers, there is a discussion of embryos being able to catch up.

Mouse: the cardia bifida shown in Figure 3 is mild compared to other cardia bifida mutants, is this also accompanied by ventral folding morphogenesis defects and do the embryos develop outside of the yolk sac?

---

## [Author Response]

*Essential revisions:*

*1) The concept of paracrine signaling should be further tested by overexpressing pdgfaa specifically in the endoderm. In addition, pdgfc (another potential ligand for Pdgfra) expression should be analyzed at the relevant stages.*

*[The endodermal-myocardial connection should be further developed, also as it is the title of the manuscript.]*

We appreciate the reviewers' interest in our proposed model in which PDGF ligands, produced by the anterior endoderm, signal through the Pdgfra receptor in the ALPM in order to control the directionality of cardiomyocyte behavior. In support of this model, our manuscript shows that *pdgfaa*, encoding one potential ligand for Pdgfra, is expressed in the anterior endoderm (Figure 7), and the reviewers wonder where other potential ligands for Pdgfra are expressed during the relevant timeframe. In accordance with this suggestion, we have examined the expression of the two other zebrafish genes encoding potential Pdgfra ligands, *pdgfab* and *pdgfc* (Eberhart, et al., 2008, Nat. Genet. 40:290). We found that *pdgfab* is expressed similarly to *pdgfaa* in bilateral medial stripes consistent with the location of the anterior endoderm. However, we did not observe expression of *pdgfc* over background levels in this region. Therefore, it is feasible that both *pdgfaa* and/or *pdgfab* could be produced by the endoderm to signal through Pdgfra in the ALPM. In our revised manuscript, these data are presented in a new figure supplement (Figure 7—figure supplement 1) and are incorporated into our revised Results section (subsection “The Pdgfra ligand *pdgfaa* is expressed in the anterior endoderm, adjacent to the ALPM”, first paragraph).

As another type of evidence in support of our model, our manuscript shows that global overexpression of *pdgfaa* disrupts cardiac fusion (Figure 7), suggesting that the spatial restriction of *pdgfa* genes is important for regulating cardiomyocyte movement. The reviewers wonder what would happen if we overexpressed *pdgfaa* specifically in the endoderm: would this have a different effect from that caused by global overexpression? As suggested, we attempted to overexpress *pdgfaa* specifically in the endoderm and evaluate the effects on cardiac fusion. For this purpose, we generated a new transgene, *Tg(sox17:pdgfaa-2A-mCherry)*, in which *pdgfaa* expression is controlled by the endoderm-specific *sox17* promoter (Mizoguchi et al., 2008, *Development***135**:2521). Injection of the *Tg(sox17:pdgfaa-2A-mCherry)* plasmid induced the mosaic expression of *pdgfaa* within the endoderm, as confirmed by examining the colocalization of GFP and mCherry after injection into embryos carrying *Tg(sox17:egfp)* (data not shown). By analyzing expression of the myocardial marker *myl7* at 22 s, we found that injected embryos had significant defects in cardiac fusion, including cardia bifida (Figure 8). In addition, the bilateral populations of cardiomyocytes in injected embryos appeared highly dysmorphic and were often accompanied by ectopic *myl7* expression (Figure 8). These phenotypes are distinct from those observed in *pdgfra* mutants (Figure 1) or those induced by global overexpression of *pdgfaa* at the tailbud stage (Figure 7). Moreover, these phenotypes suggested that expression of *Tg(sox17:pdgfaa-2A-mCherry)* might disrupt the formation of the ALPM in the early gastrula. Indeed, we found that embryos injected with *Tg(sox17:pdgfaa-2A-mCherry)* displayed dysmorphic and ectopic expression of the ALPM marker *hand2* at 6 s (Figure 8). In addition, we found that embryos with particularly high mosaicism of *Tg(sox17:pdgfaa-2A-mCherry)* exhibited morphologically evident gastrulation defects (data not shown). These early defects in gastrulation and ALPM formation correspond with the early expression of *sox17,* which initiates in endoderm cells during gastrulation (Kikuchi et al., 2000, *Genes Dev.***14**:1279). These results are also consistent with previous studies in frog and chick that have shown a role for PDGF signaling in regulating mesoderm movement during gastrulation (Nagel et al., 2004, *Development***131**:2727; Yang et al., 2008, *Development***135**:3521; Damm et al., 2011, *Development***138**:565).

Author response image 1.Mosaic expression of *pdgfaa* in the endoderm causes defects in cardiac fusion and ALPM formation.(**A-D**) Lateral (**A**, **B**) and dorsal (**C**, **D**) views depict the expression of *myl7* at 22 s in a representative uninjected embryo (wt; **A**, **C**) and in a representative embryo injected with *Tg(sox17:pdgfaa-2A-mCherry)* plasmid (injected; **B**, **D**). Injected embryos exhibited a range of cardiac fusion phenotypes: cardia bifida with disrupted morphology (**B**, **D**; n=6/14), fusion only in posterior positions (not shown; n=3/14), or normal fusion (n=5/14). (**E-H**) Lateral (**E**, **F**) and dorsal (**G**, **H**) views depict the expression of *hand2* at 6 s in a representative uninjected embryo (wt; **E**, **G**) and in a representative embryo injected with *Tg(sox17:pdgfaa-2A-mCherry)* plasmid (injected; **F**, **H**). Instead of being confined to narrow bilateral stripes of ALPM as in wt (**E**, **G**), *hand2* was expressed ectopically in some injected embryos (F, H; n=2/11) or expressed in a dysmorphic ALPM (not shown; n=5/11). Normal *hand2* expression was observed in a few injected embryos (n=4/11). Scale bars: 60 µm.**DOI:**
http://dx.doi.org/10.7554/eLife.21172.026

Unfortunately, the early defects caused by expression of *Tg(sox17:pdgfaa-2A-mCherry)* preclude the assessment of cardiac fusion in this context. If gastrulation and ALPM formation are abnormal, we cannot effectively evaluate the subsequent progression of cardiomyocyte movement toward the midline. In future studies, we plan to develop more sophisticated tools to control the expression of *pdgfaa* both temporally and spatially. For example, we plan to construct new transgenes and new stable lines employing the HOTcre system (Hesselson et al., 2009, *PNAS***106**:14896) in order to regulate *pdgfaa* expression in the endoderm after gastrulation is complete. Ideally, we aim to generate the appropriate strains for the conditional inactivation of *pdgfaa, pdgfab*, and *pdgfra* in a tissue-specific and temporally-controlled manner, since this would be a particularly effective way to test whether PDGF ligands produced by the endoderm signal through Pdgfra in the ALPM in order to influence the direction of myocardial movement. We have incorporated this idea into our revised Discussion section (first paragraph). Since the development of the necessary tools for these experiments will require more time than is appropriate for a standard revision, we view this as a separate body of work that lies beyond the scope of our current manuscript.

We hope that these modifications to our manuscript will satisfy the reviewers' interest in further development of the endoderm-myocardium connection. We note that the reviewers commented that this connection "is the title of our manuscript". We presume that they meant to point out that we mention this connection in our Abstract, since our title did not include any reference to the endoderm.

*2) The penetrance of the phenotypes is hard to gauge for both the fish and mouse data. This would be best represented in a table, instead of the figure legends (Figure 1—figure supplement 1 and Figure 3), where some (but not all) of the data is present. See below:*

Fish: The authors must have genotyped all of the embryos, how many of these are "morphologically evident mutants"? It is important to provide this information as this may significantly reduce the incidence of the mutants.

Fish, if the authors only see four cardia bifida mutants, and the incidence is also low in the mouse, mention of this phenotype should be removed from the abstract as this is misleading (not the dominant phenotype).

Mouse: can some of the defects in heart development simply be a delay? In the old Grüneberg papers, there is a discussion of embryos being able to catch up.

*Mouse: the cardia bifida shown in Figure 3' is mild compared to other cardia bifida mutants, is this also accompanied by ventral folding morphogenesis defects and do the embryos develop outside of the yolk sac?*

The cardiac phenotypes in *pdgfra* mutant zebrafish and *Pdgfra* mutant mice are incompletely penetrant and have variable expressivity. As suggested by the reviewers, we have added four new tables to our revised manuscript to clarify these points (Table 1, Table 2, Table 4, and Table 5). We have also adapted our revised text (Results) and figure legends (Figure 1—figure supplement 1 and Figure 1—figure supplement 2; Figure 3) to accommodate the incorporation of these new tables. The data presented in Table 2 illustrate the frequency with which we typically observed morphologically evident cardiac phenotypes in zebrafish *ref* mutants: that is, the penetrance of the *ref* mutant phenotype appeared higher when we examined embryos at earlier stages (approximately 75% penetrance at 20 s) than it did at later stages (approximately 44% penetrance at 48 hpf). These data suggest that some *ref* mutants recover as development proceeds, and we have added this point to our revised text (Figure 1 legend). Finally, as suggested by the reviewers and in accordance with the data shown in our new tables, we have removed mention of cardia bifida from the Abstract.

We have also revised our text to address the reviewers' questions about the cardiac defects observed in *Pdgfra* mutant mice. We agree with the reviewers that developmental delay is an important consideration when evaluating cardiac phenotypes. However, the severe cardiac defects that we observed in *Pdgfra* mutants (Figure 3) are not likely to be a result of general developmental delay, as these mutant phenotypes do not resemble wild-type cardiac morphology at younger stages. We had included this point in our original manuscript within the legend to Figure 3; to highlight it more effectively, we moved this information into our revised Results section (subsection “Mutation of *Pdgfra* disrupts heart tube assembly in mice”, last paragraph). The reviewers also wondered whether the cardia bifida that we observed was accompanied by ventral folding morphogenesis defects or by development outside of the yolk sac. We did not observe these features, and we have incorporated this information into our revised Results section (see aforementioned subsection).